



# Getting the leaves right matters for estimating temperature extremes

Gregory Duveiller[1,2], Mark Pickering[3], Joaquin Muñoz-Sabater[4], Luca Caporaso[2], Souhail Boussetta[4], Gianpaolo Balsamo[4], and Alessandro Cescatti[2]

[1]Max Planck Institute for Biogeochemistry, Jena, Germany
[2]European Commission Joint Research Centre, Ispra, Italy
[3]JRC consultant, Ispra, Italy
[4]European Centre for Medium Range Weather Forecasts, Reading, UK

**Correspondence:** Gregory Duveiller (gduveiller@bgc-jena.mpg.de)

**Abstract.**

Atmospheric reanalyses combine observations and models through data assimilation techniques to provide spatio-temporally continuous fields of key surface variables. They can do so for extended historical periods whilst ensuring a coherent representation of the main Earth system cycles. ERA5, and its enhanced land surface component ERA5-Land, are widely used in Earth System science and form the flagship products of the Copernicus Climate Change Service (C3S) of the European Commission. Such land surface modelling frameworks generally rely on a state variable called leaf area index (LAI), representing the amount of leaves in a grid cell at a given time, to quantify the fluxes of carbon, water and energy between the vegetation and the atmosphere. However, the LAI within the modelling framework behind ERA5 and ERA5-Land is prescribed as a climatological seasonal cycle, neglecting any inter-annual variability and the potential consequences that this uncoupling between vegetation and atmosphere may have on the surface energy balance and the climate. To evaluate the impact of this mismatch in LAI, we analyse the corresponding effect it has on land surface temperature (LST) by comparing what is simulated to satellite observations. We characterise a hysteretic behaviour between LST biases and LAI biases that evolves differently along the year depending on the background climate. We further analyse their repercussion on the reconstructed climate during the more extreme conditions in terms of LAI deviations, with a specific focus on the 2003, 2010 and 2018 heatwaves in Europe where LST mismatches are exacerbated. We anticipate that our results will assist users of ERA5 and ERA5-Land data to understand where and when the larger discrepancies can be expected, but also guide developers towards improving the modelling framework. Finally, this study could provide a blueprint for a wider benchmarking framework for land surface model evaluation that exploits the capacity of LST to integrate the effects of both radiative and non-radiative processes affecting the surface energy.

## 1 Introduction

The state of the land surface modulates the exchange of water and energy between the land and the atmosphere (Seneviratne et al., 2010). It can thus affect the physical state of the atmosphere and therefore influence the seasonal to inter-seasonal predictability and climate projections (Koster et al., 2004). The biophysical land-atmosphere interactions are determined by land surface properties, such as albedo, emissivity, surface roughness and evaporation (Anderson-Teixeira et al., 2012), all of



which can be highly heterogeneous in both space and time (Santanello Jr et al., 2018). As a result, the partition of available
energy into latent and sensible heat fluxes can be highly variable over emerged surfaces of the planet (Dickinson, 1995). The
result of this allocation has a direct impact on local surface or near-surface air temperature (Pielke Sr et al., 2002), which in
turn can exacerbate the impacts of anthropogenic climate change.

The type and density of vegetation covering the land surface have a strong role in determining the surface energy balance.
Land cover is normally classified into broad groups summarising land surface properties involved in land-climate interactions.
Under similar conditions of radiation, a forest will generally absorb more energy than low vegetation (i.e. grasses or crops) due
to its darker surface but, in terms of surface temperature, this is generally more than compensated by the larger amount of energy
released back to the atmosphere through higher transpiration, which itself is possible due to improved access to water through
deeper roots (Bonan, 2008). Differences in land cover have been shown to affect land surface temperature (LST) (Duveiller
et al., 2018; Alkama and Cescatti, 2016; Li et al., 2015) and even affect the cloud regime above them (Duveiller et al., 2021;
Xu et al., 2022). However, land surface characteristics also vary at different time-scales within similar land cover classes and
are further affected by both natural processes and land management (Anderson et al., 2011). Particularly in extra-tropical
regions, land characteristics exhibit strong seasonal patterns due to the cycle of leaf development and senescence, influencing
the seasonality of albedo, surface roughness length and fluxes of water and energy (Richardson et al., 2013). Another way to
characterise the overall state of the vegetated land that more readily catches such differences in a spatially continuous way
is to consider the state variable known as Leaf Area Index (LAI). LAI is defined as half of the total green leaf area per unit
horizontal ground surface area (Yan et al., 2019). It is a key variable as it represents the exchange surface between plants and
the atmosphere, at the intersection between water, energy and carbon cycles, thus playing a critical role in the feedback of
vegetation to the climate system (Fang et al., 2019; Forzieri et al., 2017). LAI exhibits a large seasonal variability accordingly
to climate zones and vegetation types and a substantial inter-annual variability linked to year-to-year variability in weather or
in management (Boussetta et al., 2015). To a large extent, LAI drives the temporal changes in biophysical properties within a
given land cover type, since some properties, such as albedo and stomatal conductance, can still differ among vegetation types
for the same value of LAI due for instance to morphological differences in leaf types (e.g. broadleaf versus needleleaf). In any
case, in any effort to estimate or monitor land-atmosphere interactions and their consequences, getting the quantities of leaves
right seems to be an important consideration for climate reconstruction and prediction.

The impacts of land-atmosphere interactions on local temperature are exacerbated during extreme events such as heat-
waves (Jia et al., 2019). The trigger of such events is often an atmospheric circulation anomaly governed by persistent anticy-
clones (Schubert et al., 2014; Brunner, 2018), enabling cloud-free conditions and an increase of net solar radiation.  Heatwaves
can be locally intensified by land-atmosphere feedbacks, which in turn may result in enhanced growth of the atmospheric
boundary layer that increases the entrainment of heat (Miralles et al., 2011) and/or horizontal heat advection (Schumacher
et al., 2019). In addition, a deficit in the soil moisture content can further warm the air (Hauser et al., 2016)  so that both
thermodynamic and dynamic drivers could act synergically (Coumou et al., 2018), leading to an amplification of major heat-
waves (Horton et al., 2016). As a result, heatwaves often occur as compound events characterised by a persistent drought
that increases the intensity of the heatwave (e.g. Miralles et al., 2012; Seneviratne et al., 2010). Studies suggest that dense





vegetation can limit the amplitude of heat extremes (Renaud and Rebetez, 2009), with deciduous and mixed forests having
a stronger cooling effect compared to conifer forests. Understanding the role of vegetation states on these phenomena is becoming increasingly relevant as heatwaves have increased in intensity, frequency and duration (Perkins-Kirkpatrick and Lewis, 2020), with these trends getting worse as the climate warms up (Christidis, 2015; Coumou et al., 2018) due to various factors such as the increased climate variability (Schär et al., 2004), a weakening of soil moisture constraints (Rasmijn et al., 2018) and reduced plant transpiration due to $CO_2$ physiological forcing (Skinner et al., 2018). In addition, observation data reveal a
stronger increase of high temperatures over land compared to trends in global mean temperature, and this is particularly true for the most extreme events (Seneviratne et al., 2014). The relevance and impact of land atmosphere interactions is also likely to extend to more northern regions, as demonstrated in a recent study on heatwaves over northern Europe (Dirmeyer et al., 2021).

To monitor the changing state of the Earth System, including heatwaves, it is essential to have reliable data that is spa-
tially and temporally consistent and modelling frames that mechanistically represent the interplay between the key variables. Although the availability of Earth Observation (EO) data has been increasing in terms of quality, quantity and diversity, they remain constrained by two main issues: (1) EO records can have spatio-temporal gaps and (2) several state variables can simply not be measured directly. These shortcomings can be compensated by integrating observations within a modelling framework, which is where reanalysis comes into play. By optimally combining observations and models through data assimilation tech-
niques, reanalyses can provide spatio-temporally continuous fields of variables for an extended historical period while ensuring the integrity and coherence in the representation of the main Earth system cycles (Hersbach et al., 2020; Dee et al., 2011).

One of the most widely-used reanalysis for Earth System Science is the atmospherical reanalysis of the European Centre for Medium-Range Weather Forecasts (ECMWF). Currently, the latest installment of this dataset is the fifth generation of atmospheric reanalysis called ERA5 (Hersbach et al., 2020), and it is produced using 4D-Var data assimilation and a ECMWF model
forecast (the Integrated Forecast System (IFS)) version corresponding to the ECMWF cycle cy41r2. Within the IFS, an atmospheric model is coupled both to an ocean model and to a land-surface model, the latter being responsible for correctly representing the land-atmosphere interactions introduced above. This model was originally called TESSEL, for Tiled ECMWF Scheme for Surface Exchanges over Land. It was revised to address shortcomings of the land surface scheme to represent the hydrology to become HTESSEL (Balsamo et al., 2009). An additional land surface CO2 exchange module was added to
enable environmental forecasting applications which also involves interaction with atmospheric CO2 concentration, leading to CHTESSEL (Boussetta et al., 2013). The land surface model has more recently evolved into ECLand, a modular system that should facilitate modular extensions for the benefit of efficient developments and external collaborations (Boussetta et al., 2021).

ERA5 is now a flagship product of the European Commission's Copernicus Climate Change Service (C3S) and is widely
used across diverse fields. Within C3S, ECMWF has also produced an enhanced land component of ERA5, known as ERA5-Land (Muñoz-Sabater et al., 2021). It is produced by re-running the land component of the ERA5 reanalysis at a finer spatial resolution driven by the original atmospheric forcing from ERA5. This results in land variables at a higher horizontal resolution ($\approx$ 9 km) than those available from ERA5 ($\approx$ 31 km). It is also cost-effective way to produce very consistent land variables over



several decades, as observations are not directly assimilated and the land component is not coupled to the atmospheric or ocean

model. The fact that both ERA5 and ERA5-Land are now an integral and operational part of C3S means that their production is guaranteed with timely updates. Furthermore, following the efforts of the CO2 Human Emissions (CHE) project (Balsamo et al., 2021), ECLand should become the engine of the prototype Copernicus CO2 monitoring tool within the follow-up CoCO2 project (https://coco2-project.eu/). Given its prominent role in all these initiatives, there is a high interest in further evaluation the capacity of ECLand within the ERA5 and ERA5-Land framework to correctly represent land-atmosphere interactions, in

particular under the extreme conditions of heatwaves.

In order to correctly characterise land-atmosphere interactions, a variable that a modelling system should ideally predict accurately is LST, as it governs the interface between water and energy fluxes. Several studies have revealed how the land model behind ERA5 and ERA5-Land suffers from a strong bias in its representation of LST (Johannsen et al., 2019; Nogueira et al., 2020; Orth et al., 2017). They all conclude that incorrect descriptions of the vegetation are largely responsible for such

poor model performances. Orth et al. (2017) demonstrated that there is no region across Europe or Africa where both mean LST or its seasonal dynamics are well captured by the CHTESSEL model, but they also suggest that considerable improvement can be gained by calibrating with multiple observation-driven datasets. Focussing on the Iberian Peninsula, Johannsen et al. (2019) found that replacing the land cover representation with a newer ESA-CCI map could reduce the summer bias. Nogueira et al. (2020) confirmed that this LST bias problem with CHTESSEL was also present in the widely used ERA5 data. They

further showed how another land surface model (SURFEX-ISBA) did not display the cold bias over Iberia and attributed this improvement to both a better land cover description and a more appropriate seasonal evolution of LAI, including a clumping parametrization for low vegetation. Based on these results, Nogueira et al. (2021) updated both land cover and vegetation seasonality in the ECMWF coupled system to show the potential of reducing the LST bias beyond Iberia. This work however highlights the complex regional heterogeneity in the atmospheric sensitivity to land cover and vegetation changes, calling for

re-calibration of the model parameters and re-evaluation of model assumptions for future reanalyses.

Although the misrepresentation of vegetation types has clearly been identified as a main culprit in the shortcomings of LST representation in ERA5 and ERA5-Land, there is still a main issue that has yet to be investigated: LAI dynamics. In both ERA5 and ERA5-Land, LAI is always prescribed at grid cell level with an identical seasonal cycle based on satellite-derived LAI (Boussetta et al., 2012). While this had been considered as an improvement compared to a more basic look-up table

approach employed in the past (Boussetta et al., 2012), it still neglects any inter-annual variability in the phenology and density of vegetation. This means that any year where a variation is observed from this climatological seasonal cycle, in terms of either phase or amplitude, will lead to a discrepancy between reality and the land representation in the modelling framework. Such mismatch may be particularly exacerbated in situations of heatwaves, as plant phenology has been shown to vary substantially under dry/hot extremes and have important impacts on the development of these events (Stéfanon et al., 2012; Skinner et al.,

2018; Lorenz et al., 2013). The benchmarking exercises mentioned before (Johannsen et al., 2019; Nogueira et al., 2020, 2021) only considered changes in the sources of the LAI products, but always kept it as a prescribed seasonal cycle, leaving no room to explore the dynamical nature of the LST bias with respect to the LAI mismatch.





In this paper, we propose an alternative take on evaluating the importance of LAI variations on the LST biases within the modelling framework that produces the ERA5 and ERA5-Land datasets. We focus specifically on the dynamic nature of these biases and their possible repercussion on the accuracy in the reconstruction of heatwaves. The objective of this work is two-fold: (1) to make a comprehensive diagnostic of how the combined biases in LAI and LST evolve in space and time and across climate zones; (2) to evaluate the repercussion this has on our capacity to represent heatwaves over Europe and set the basis for a future improvement of the system.

## 2 Material and methods

### 2.1 Reanalysis data

The main data used in this study is the reanalysis data. All of it is available from the Copernicus Data Store (CDS) of the C3S service (https://cds.climate.copernicus.eu/). The priority is to investigate the data in ERA5-Land, as users interested in land and land-atmosphere interactions would probably opt for the dedicated land product at the finer spatial resolution of 0.1° rather than ERA5. Besides, ERA5-Land shows very good consistency in the longer time records, whereas ERA5 surface variables with long memory present frequent inconsistencies (Muñoz-Sabater et al., 2021). However, some variables needed for the study are only available in ERA5. Therefore, the entire study is focused on the 0.25° grid of ERA5, and all variables used, whether from ERA5-Land or from satellite products, are aggregated back to the 0.25° grid. To avoid any confusion and to remind the reader that the underlying data is mostly pertinent to ERA5-Land, we will henceforth use the acronym ERA5L to refer to the dataset prepared in this study, while reserving ERA5 and ERA5-Land to design the original data sources. The time period considered for ERA5L ranges from 2003 until 2018.

Each variable from reanalysis needs to be matched with a respective equivalent from satellite-derived products that serves as an 'observational' reference. For most of these variables, there are generally various different sources of satellite products to choose from. The choices we made were guided by the aim to use products that are independent from the ERA5, ERA5-Land and C3S environments. For each variable, the match between the satellite reference and the reanalysis variables requires some specific considerations, and these will be discussed on a per-variable basis in the following subsections.

### 2.2 Leaf Area Index

The satellite-derived LAI product that we use in this study is GEOV2/AVHRR (Verger et al. 2021). This product is based on applying a neural network retrieval algorithm on the AVHRR Long Term Data Record (LTDR, version 4, available at https://ltdr.modaps.eosdis.nasa.gov/cgi-bin/ltdr/ltdrPage.cgi). Additionally, this product benefits from a thorough pre-retrieval spectral harmonization and a post-retrieval gap-filling procedure. This product was designed to have a high consistency with the GEOV2-CGLS products derived from VEGETATION and PROBA-V sensors, distributed by the Copernicus Global Land Service (CGLS), and which have been found to improve the LST bias in previous studies (Nogueira et al., 2020, 2021). The original product is provided at 0.05° spatial resolution with a 10-daily timestep, and it is aggregated to month-





ly values at 0.25° to match the ERA5L LAI. From the reanalysis side, the prescribed LAI is obtained from the ERA5 monthly
averaged data on single levels. It is obtained by combining the variables called *Leaf area index, high vegetation* and *Leaf area index, low vegetation* on a per-grid cell basis using the fractions of high and low vegetation prescribed in the model, known respectively as *High vegetation cover* and *Low vegetation cover*.

## 2.3   Land surface temperature

The observational reference for LST is obtained from the Moderate Resolution Imaging Spectroradiometer (MODIS) instrument on-board of the Aqua satellite platform. MODIS-Aqua was selected as its overpass time is approximately 13:30 local time, which would be close to the time of the daily maximum temperature. The precise MODIS data product is labeled as MYD11A1 collection 6 (Wan et al., 2015), based on a split-window algorithm, and provides data at 1km spatial resolution at a daily frequency. The variable in ERA5-Land that we compare LST to is called "*skin temperature*" and is defined as the theoretical temperature that is required to satisfy the surface energy balance. We select skin temperature at 14:00 so as to match as close as possible the overpass time of the MODIS-Aqua instrument.

To match the reanalysis variable with remote sensing observations, special care is needed to address the clear-sky bias. The type of thermal satellite data that we use can only provide an information on the temperature's surface in the absence of clouds, which typically leads to sampling the warmer days benefiting from unobstructed solar radiation. The reanalysis dataset contains values for both sunny and overcasts days, when the skin temperature is closer to air temperature. To ensure comparability and have information at monthly scale, only the 5 warmest days of each month are selected from both the MYD11A1 and the ERA5-Land datasets. To facilitate processing for the satellite data, this procedure is directly implemented in the Google Earth Engine (GEE) platform (Gorelick et al., 2017), which hosts a copy of the MYD11A1 catalogue. The aggregation to the 0.25° grid is done in a second step. As a consequence of this matching procedure, the LST bias is always referring to a bias in the five warmest days of the month. This assumes that the five warmest days are clear sky, but this assumption should generally hold as we are using LST at around 14:00, which is a variable that is highly sensitive to radiation.

## 2.4   Albedo

While the focus of this work is on the relationship between the LAI and LST biases, it is also useful to investigate how biases in relevant biophysical variables could help the mechanistic interpretation of the discrepancies between LAI and LST in the modelling framework. The first of these variables is albedo, the proportion of the incident solar radiation that is reflected by the surface. It can serve as an indicator of whether the LST bias in ERA5L is caused by the radiative effect of changes in LAI, as an increase in LAI reduces albedo, which in turn increases net radiation leading to a radiative warming of the surface.

The variable we use in ERA5 is "*UV visible albedo for diffuse radiation*", which is the fraction of diffuse solar (shortwave) radiation with wavelengths between 0.3 and 0.7 $\mu$m reflected by snow-free land surfaces. Like LAI, albedo in ERA5 is not dynamic, but consists instead of a static seasonal climatology. From the satellite-based perspective, we use the standard MODIS daily albedo product, MCD43C3 V006 (Schaaf and Wang, 2015). From this dataset, we use the broadband white-sky estimations for the visible part of the spectrum, defined for this product as ranging from 0.4 to 0.7 $\mu$m. This means some





slight discrepancy with ERA5 might occur for ultraviolet light over the 0.3 to 0.4 $\mu$m range, but this is expected to be very marginal due to the low contribution of UV light to ecosystem scale albedo. There is a second discrepancy with ERA5 in that this MODIS albedo product is not snow-free, but this should not be a problem as albedo is only used here in the context of
studying summer heatwaves were only minimal snow cover is expected over the mountain ranges.

## 2.5 Land evaporation

The second associated variable is land evaporation. This is the amount of water that is evaporated from the land surface, including the transpiration from vegetation, and which is transformed into water vapour in the air above. In ERA5-Land, the variable is called "*Total evaporation*" and is provided as meters of water equivalent at a monthly basis. Evaporation or tran-
spiration cannot be directly measured from satellite observations, but a data-driven estimation can be obtained from dedicated modelling frameworks. The one we use here is the Global Land Evaporation Amsterdam Model (GLEAM) product (Martens et al., 2017; Miralles et al., 2011). The GLEAM model estimates land evaporation and its components: transpiration, bare-soil evaporation, interception loss, open-water evaporation and sublimation. The version used here is version 3.3b, which does not rely on ERA5 reanalysis data in order to avoid any circularity in our benchmarking work.

## 2.6 Climate zones

As land-atmosphere interactions are often related to the background climate regime, it is often useful to stratify their analysis along with some kind of climate zonation. Here we employ two different climate classification approaches. The first consists in using the well-defined Köppen-Geiger classification scheme, as implemented for the period 1986-2010 by (Kottek et al., 2006). The scheme defines five broad groups: equatorial, arid, temperate, continental and polar, as well as subgroups depending
on the seasonal rainfall and temperature. The maps are aggregated from their native spatial resolution of 1 km to the 0.25° grid using the nearest neighbour interpolation. To have a finer division of climate along continuous axes of temperature and aridity, a second climate zonation is done based on a division of the world using intervals of yearly-averaged 2m air temperature and yearly-averaged soil moisture. For this general purpose of climate characterisation, these variables are collected from the monthly ERA5-Land single layer dataset.

## 2.7 Heatwaves

To isolate the specific effect of the interplay between the LAI and LST biases during extreme events, this study also looks at three major summer European heatwaves that have occurred in the recent past. All three are characterised by a long duration and large-scale extent, but varied in terms of geographic distribution and biomes affected. The first is the heatwave of summer 2003 that hit western Europe, and particularly France, and which will be here referred to as HW03. The second is the Russian
heatwave of July 2010, referred to henceforth as HW10. The third heatwave considered occurred in 2018 and can be divided into two zones where the effects had marked differences notably due to contrasting land cover and background climate. The first zone we consider is labelled HW18a and covers Northern Germany and Denmark, a region dominated by croplands, while





the second is located mostly over forests in Finland and is labelled HW18b. The spatial extents of the zones considered are

represented in Fig. 1, overlaid over the LST anomalies from the period 2003-2018 for the respective months considered for

each event. On this point, we underline the fact that the characterisation of the heatwaves is done based on monthly data to

remain consistent with the rest of the analyses, despite the fact that heatwaves would be more precisely defined by considering

their duration more precisely at a daily scale.

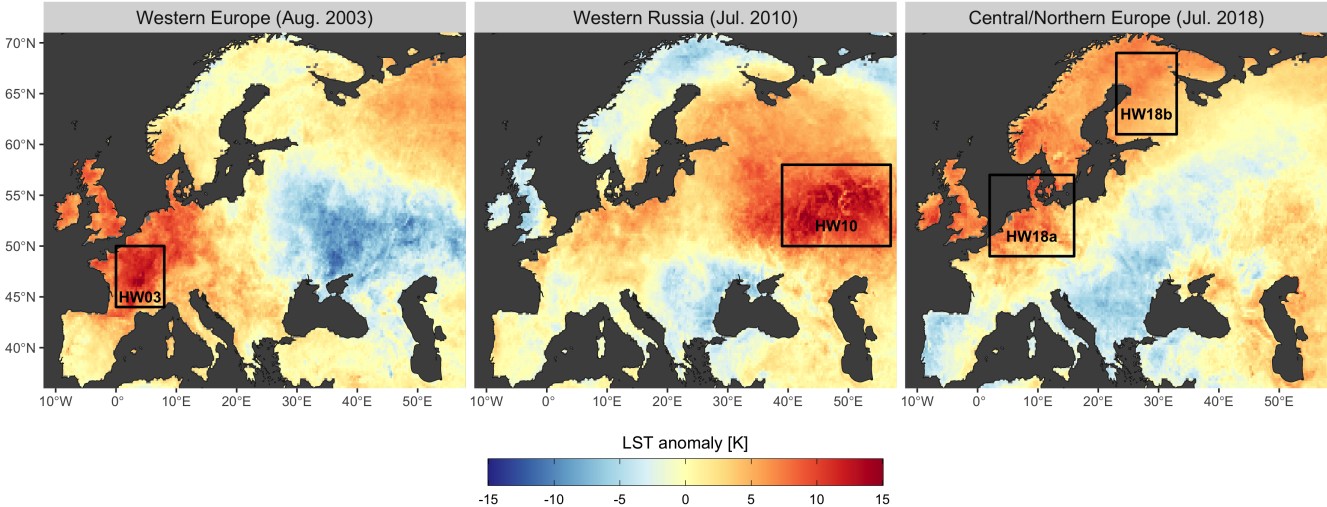

**Figure 1.** Delimitations of the areas considered for the various heatwave events considered in this study. The LST anomalies presented are

based on satellite retrievals from MODIS.

## 3 Results

The outcomes of this study are all based on the analysis of biases between ERA5L and other observational datasets for the

specific variables of LAI and LST. The results are structured along the two main objectives mentioned before. The first part

thus characterises the general behaviour of how these biases interact based on their climatologies, here considered as their

mean cycles over the period 2003-2018. The second part then takes a more specific look on how these biases interact during

years that are different from this inter-annual mean cycle, and more particularly on how this affects years of heatwaves.

**Part 1: Characterisation of the patterns based on the climatology**

To begin, we first start with a general overview of how the biases in LAI and LST are structured in space and time. The maps

in Fig. 2 are composite images where winter is represented by values for January in the Northern Hemisphere and values for

July in the Southern Hemisphere, while the reverse is true for summer. The LAI in ERA5L is almost systematically higher





than the reference GEOV2-AVHRR LAI during winter, and this corresponds to an overestimation of LST by ERA5L in the northern latitudes. This relationship between the bias in LST and the bias in LAI is consistent for such energy limited situation

where biophysical effects of vegetation on climate are dominated by radiative effects. In fact, the modelling framework assumes there is an excess of leaves covering the background, the former being generally darker than the latter (especially when the background is covered with snow) resulting in more heat accumulation than what would be observed in a situation with fewer leaves. Since the winter evapotranspiration is strongly limited by the atmospheric evaporative demand at such high latitudes, there is no compensation of the enhanced radiative warming from evaporative cooling (Bright et al., 2017). On the other

hand, there is a considerable underestimation of the LST for many drier parts of the world where winter evapotranspiration is not energy-limited. This pattern can be explained by the additional evaporative cooling generated by the excess LAI in these climate conditions. The situation is summertime is more complex, as LAI can either be overestimated or underestimated depending on the geographical location. The biases in LST generally consist of an underestimation in ERA5L with respect to the satellite reference, which is particularly strong in deserts and drylands. There is a notable exception in Central and Western

Africa where the LST is rather overestimated in ERA5L. The interpretation of how both LAI and LST biases are related is not straightforward due to the dynamic nature of this relationship along the growing season and between climate regions.

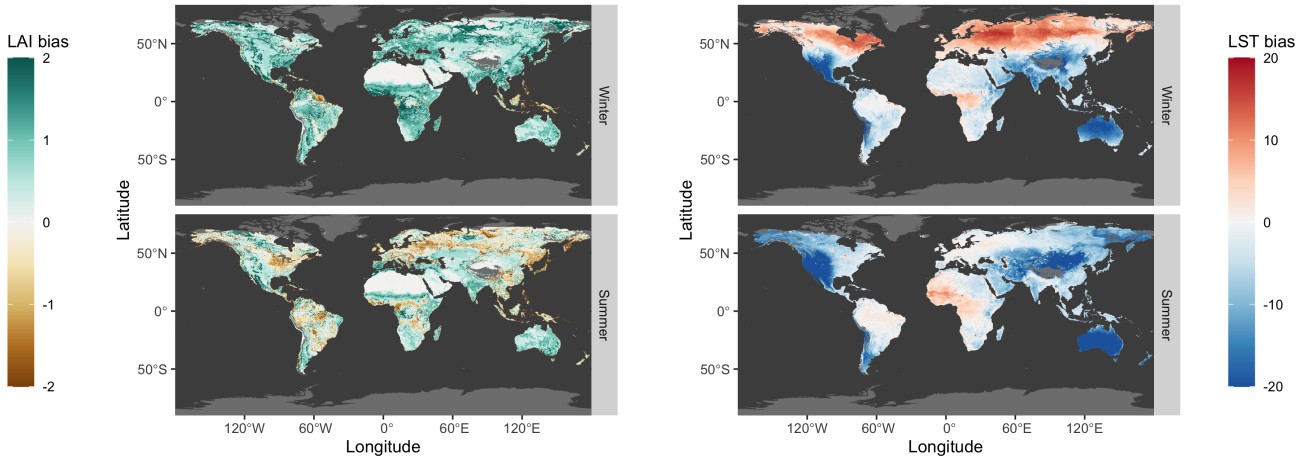

**Figure 2.** Overview of the mean biases in LAI and LST between ERA5L and observations (ERA - obs) over the climatological period from 2003 to 2018. The panels represent composite maps for which the seasonalities of both Northern and Southern Hemispheres are aligned: winter maps consist of data for January in northern latitudes and July in southern latitudes, while summer maps combine July values in the North with January values in the South.





To better diagnose the relationship between the bias in LST and the bias in LAI, we plot them once against the other to analyse their cyclic seasonal patterns as shown in Fig 3. These plots summarise the bias for all areas under a specific climatic regime defined by a range in a yearly average of soil moisture and 2m air temperature. Fig 3a represents a region in a tropical semi-humid climate, while Fig 3b is a colder and more humid climate. In both plots, a typical hysteretic pattern emerges, indicating how the relationship between the biases is depending on the background climate and changes during the course of the year. For the tropical region (Fig 3a), the relationship is consistent with a land biophysical signal dominated by evaporative cooling: an overestimation in LAI is associated with an underestimation in LST, which is more pronounced in the months of January to May, while during the year we observe a loop with smaller biases in summer than in fall. For the second region (Fig 3b), the growing season follows a stronger hysteretic pattern that even leads to an underestimation of LAI by ERA5L (as had been observed in Fig. 2), but there is a stark difference in pattern for the wintertime where a strong change in LST bias occurs independently of the bias in LAI. This pattern is consistent with the explanation provided before in which a winter overestimation of LAI by ERA5L in cold regions could lead to radiative warming due to the darker surface of dense vegetation, which is not compensated by any additional evaporative cooling due to the energy limitation of evapotranspiration in winter conditions.



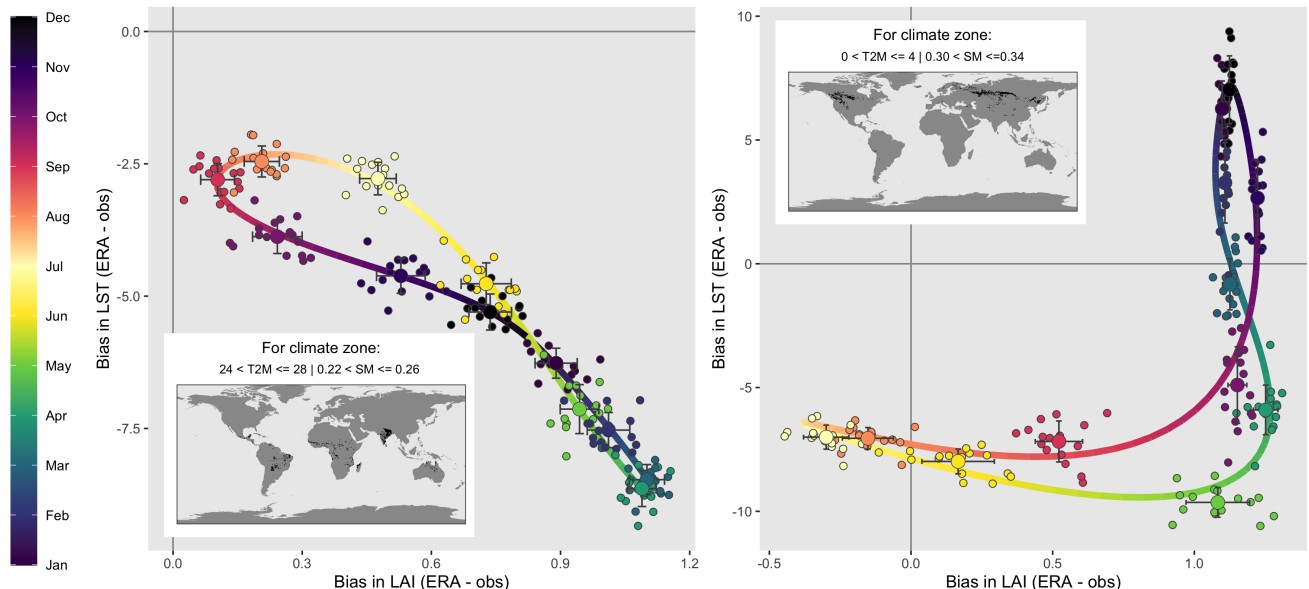

**Figure 3.** Diagnostic plots illustrating the hysteretic behaviour between the biases in LAI and the biases in LST for two different climate zones. The biases are always defined as ERA5L variables minus the values from reference observational datasets. The small individual points represent monthly values within the entire period 2003-2018. The larger points represent the inter-annual mean values for each month (and the error bars represent one standard deviation). The continuous line is obtained from a harmonic fit.

Fig 4 further extends this analysis to a full climate space delimited by mean surface soil moisture and mean air temperature. This figure exposes two general gradients in the patterns of how the LAI and LST biases behave during the seasonal cycle. The first gradient shows how the hysteretic behaviour increases for colder and moister climates, while very dry and hot regions show little variation in either bias and thereby does not show any significant hysteretic patterns. Beyond this first general gradient

on the intensity of hysteresis, there is second gradient showing a notable difference between cold and dry regions, where the magnitude of the seasonal variation in LST bias dominates, and warm and humid regions, where this seasonal variation of the bias is stronger for LAI.

In order to quantify the two main gradients described in Fig. 4, we propose two indices that respectively generalise these patterns of hysteretic intensity and bias dominance. Hysteretic intensity ($HI$) is simply summarised by the total area ($A$) formed

by the hysteretic loop in a given climate zone ($i$) divided by the area of the largest loop encountered among all climate spaces:

$$HI = \frac{A_i}{\max(A)}$$

The area is calculated based on the smoothed seasonal cycles, which themselves were fitted using 3rd order harmonics fits applied separetely to both variables as a function of time. The area A is calculated for each climate zone considering all





**Figure 4.** Hysteretic behaviour between the biases in LAI and the biases in LST for a range of climate zones defined by classes of annual mean 2m temperatures (T2m) and soil moisture (SM). The biases are always defined as ERA5L variables minus the values from reference observational datasets. Only interpolated curves are shown for clarity purposes.





intersecting loops as generating positive areas, which would not be the standard procedure from a topological perspective (as
intersecting loops would generate areas with opposite signs).

The second gradient in Fig. 4 relates to describing which of the two biases (LAI or LST) dominates in terms of seasonal
amplitude. The index to describe this behaviour follows the logic of a normalised difference index based on the standardised
ranges of both LAI and LST axes in the smoothed hysteresis curves. The resulting bias dominance (*BD*) index is expressed as
follows:

$$BD = \frac{\left( \frac{range(x)}{\sigma_x} - \frac{range(y)}{\sigma_y} \right)}{\left( \frac{range(x)}{\sigma_x} + \frac{range(y)}{\sigma_y} \right)}$$

where *x* stands for the bias in LAI and *y* is the bias in LST.

These two indices can be mapped in climate space, but then also back into geographical space, as shown in  Fig. 5. This
provides a valuable diagnostic that enables spatialisation of the magnitude of the hysteretic discrepancies between ERA5L
and observations in terms of the interrelationship between their LAI and LST biases. This in turn is useful both for users of
reanalysis data, to know where the LAI/LST land atmosphere interactions are to be expected to be problematic, and for model
developers, to know where they should prioritise model improvements. More specifically, when the HI map in Fig. 5 indicates a
dark area, one knows the relationship between biases does not have a strong seasonal component, and can instead be considered
stable. In some cases, this is because they resume to a single point (e.g. more desertic areas). In others, it is because there is
a clean quasi-linear relationship between the LAI and the LST bias (e.g. tropical forests, or the example in Fig. 3a), which
could also be empirically "corrected" using a linear fit if this was deemed appropriate or necessary for a user (although this
would compromise the physical integrity of the relationship between the variables in ERA5L).  Areas with high HI indicate
there is a strong seasonal component in the mismatch between LAI and LST biases, and this appears to affect areas with strong
seasonality in LAI and LST.



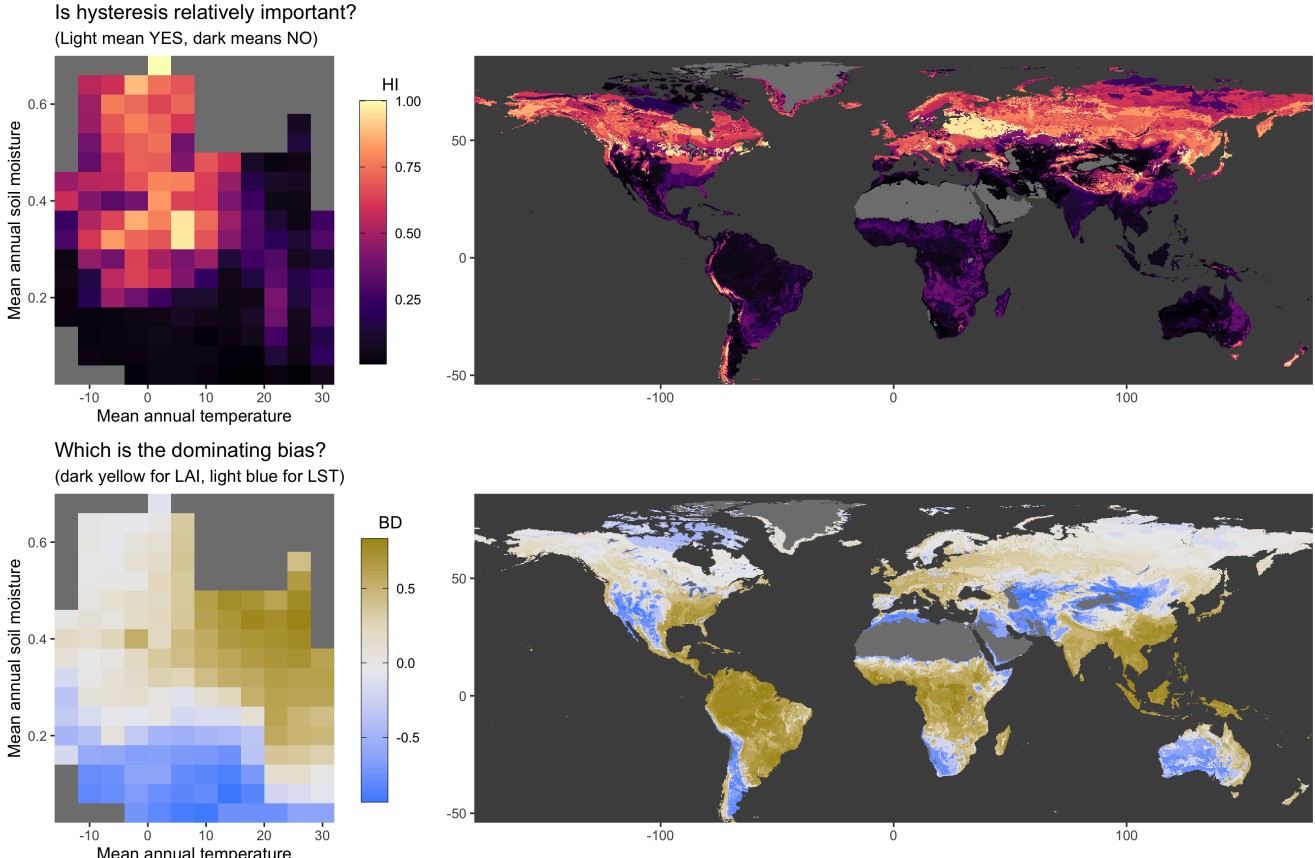

**Figure 5.** Summary of how biases in LAI and LST interact differently across climate zones (left) and how these are translated back into geographic space (right). The HI index (top) indicates how important the hysteretic patterns are. The BD index (bottom) indicates which of the two bias dominates (between LAI and LST).

**Part 2: interannual variability and heatwaves**

After characterising the general patterns of the biases based on the mean inter-annual cycles, or climatologies, we now turn the attention to extreme situations which deviate from the mean. To begin, we start by showing how the relationships between LST and LAI biases change from one year to the next. This is done via an analysis of their inter-annual variability for a specific month and place over the considered period (from 2003 to 2018). Fig. 6 displays the temporal correlation between the biases for selected months representing the seasons uniformly across the world (i.e. seasonality of the Northern and Southern Hemispheres are aligned). The most prominent patterns are negative correlations in drier areas, especially when there is strong






radiation load in summer. This comforts the previous assessment that an overestimation of LAI in the modelling framework coincides with an underestimation of the LST, but further indicates how this effect changes on a year-to-year basis. In other words, the years when the seasonally prescribed LAI of ERA5L is further from the reality, e.g. in years where the LAI peak is lower or shifted due to particular growing conditions of that year, the underestimation of LST can be expected to be more severe.

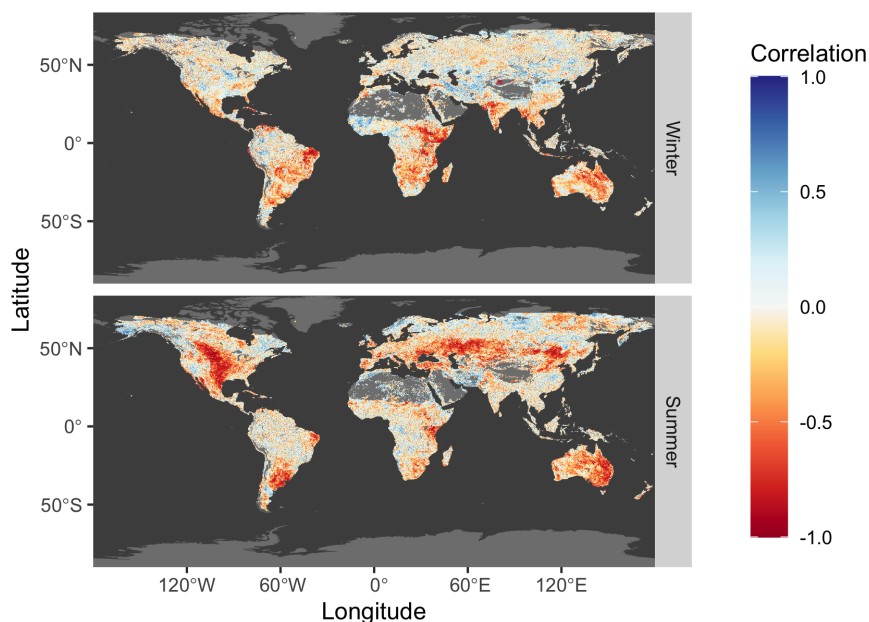

[Each season in this plot is represented by a single month, consisting of either January or July depeding on whether the data is in the Northern or Southern Hemisphere]

**Figure 6.** Inter-annual correlation between the biases in LAI and the biases in LST based on all months of July and January over the period 2003-2018. As with Fig. 2 these are composite maps for which the seasonalities of both Northern and Southern Hemispheres are aligned: winter maps consist of data for January in northern latitudes and July in southern latitudes, while summer maps combine July values in the North with January values in the South.

To better understand how the relationship between LAI and LST reacts under conditions that deviate from the normal, the next analysis concentrates on using anomalies of temperature as a grouping variable. For the scope of this analysis, the focus is placed on Europe. Fig. 7 summarises how the LAI and LST biases evolve when considering the full range of temperature anomalies encountered in our dataset across different climate zones in Europe. To construct this plot, the entire distribution of values for a given bias are considered, effectively mixing space and time. This distribution is divided in quantiles (deciles in





this case) based on their value of land surface temperature monthly anomalies. For each group of anomalies, the average bias in LST or LAI is shown. There is a clear difference across climate zones. For LAI, the bias is relatively stable irrespective of LST anomalies in subartic climate (Dfc), but it has a tendency to increase with higher LST anomalies in humid continental climate (Dfb) or in oceanic climate (Cfb), while in mediterranean climate (Csa) it actually decreases when extremes occur.

LST largely follow the opposite patterns for the warm extremes, but not necessarily for the cold ones (most notably in Dfc and Cfb).  For the specific case of heatwaves, Fig. 7 suggests that for most of continental Europe, the bias in LAI will go from an underestimation in ERA5L to an overestimation as thermal anomalies increase, and that this will considerably aggravate the discrepancies in LST.



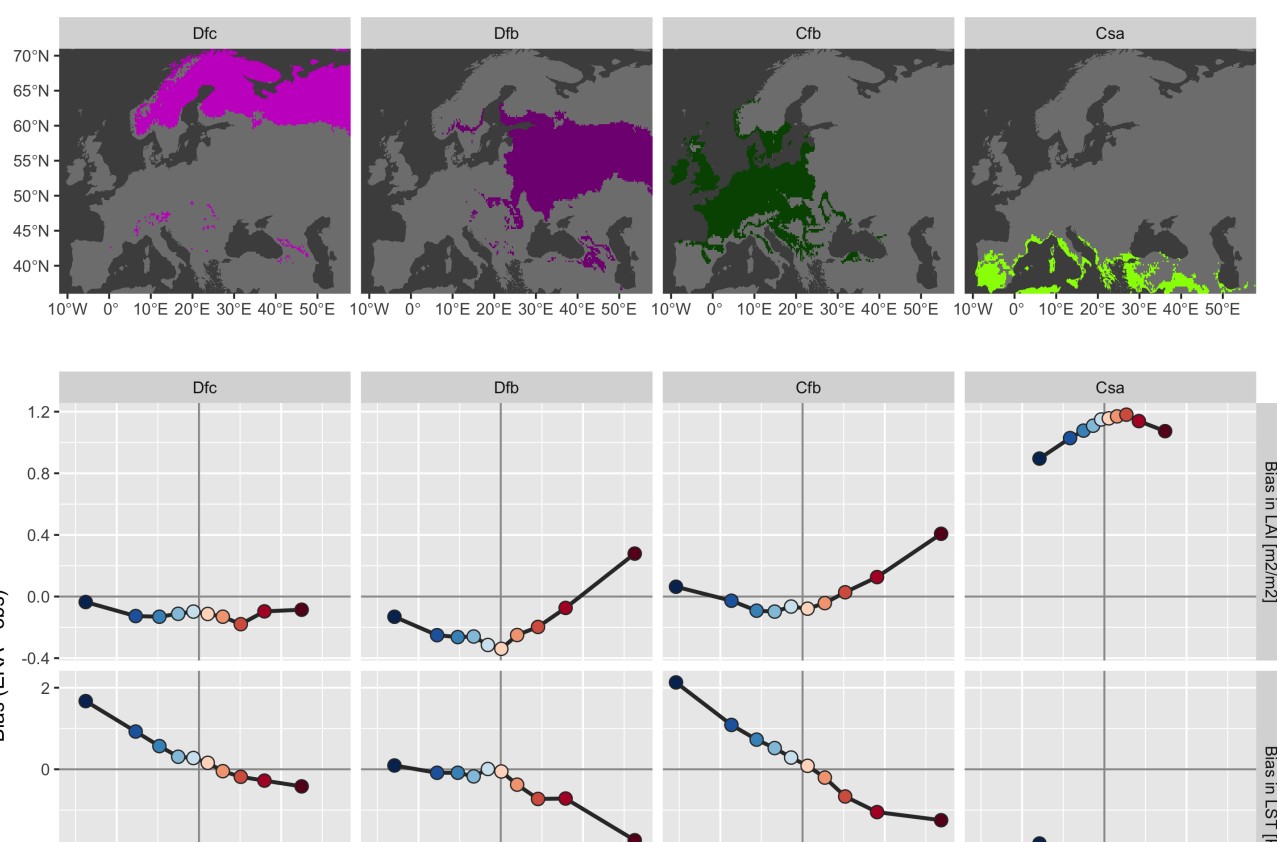

**Figure 7.** Description of how the biases in both LAI and LST (between ERA5L and observations) changes over different climate zones within Europe depending on anomaly intensities in LST.

Finally, we turn out attention to the specific case of the three major European heatwaves in 2003, 2010 and 2018 with the
latter one divided among the two regions (HW18a and HW18b). Fig. 8 maps the differences in the biases between the year of
the heatwave and the average bias for the same period, and what we refer to here as a bias shift. This bias shift only informs
us on how the bias changes from the normal year to a heatwave year, but does not indicate whether the starting situation is
an overestimation or an underestimation. Therefore, to facilitate the interpretation, Fig. 8 also includes the spatially averaged
biases for each event under the maps.





The first point to remark in Fig. 8 is that for HW03, HW10 and HW18a  there is a considerable LAI bias shift in the same direction due to the fact that LAI is effectively lower in the observed dataset during these events than in normal years. For HW03 and HW10, this changes the situation from an underestimation by ERA5L in normal years to a strong overestimation during heatwave conditions. For HW18 the situation is different, both with respect to the other heatwaves, but also among the two sub-regions considered. For the HW18a region over Northern Germany and Denmark, the LAI bias shift actually leads to

a situation in which the prescribed LAI in ERA5L is actually closer to the reduced LAI measured during that specific year, leading effectively to less underestimation than in normal situations. For the HW18b region in Finland, which is dominated by forests unlike all other considered regions, the LAI bias shift is in the opposite direction going from an overestimation of LAI by ERA5L to a slight underestimation in the heatwave year.

The second point to highlight in Fig. 8 is that all heatwave cases show a negative LST bias shift, albeit with different orders

of magnitude.  For HW03 and HW10, the LST bias shift is very strong, and it clearly corresponds to the positive shift in LAI bias. This confirms that ERA5L can suffer from a cold bias in these extreme situations, arguably attributable to an excessive evaporative cooling caused by simulating many more leaves than what is present in reality. In contrast, the spatially averaged shift in LST bias for HW18b is almost insignificant, with even some increases in some areas. This is in line with the remark that LAI over these forested areas may be better estimated during this event by the ERA5L prescribed climatology, resulting in

very little consequences to the LST bias. In the case of HW18a over northern Germany, the improvement in LAI for the event almost entirely removes the positive LST bias that exists in normal conditions.

To better understand how the biases in LST and LAI are effectively related in the contrasting heatwave circumstances, Fig. 8 also provides the same maps for the shift in two other variables:  shortwave albedo and total evaporation. The albedo maps show the shift that would be expected over cropland-dominated areas during heatwaves, i.e. the senescence of cereals

would be accelerated resulting in brighter surfaces as cereals dry off, resulting in a negative albedo shift when comparing the real observed albedo with the prescribed one. This is clearly not visible over the forested HW18b zone where the LAI bias present in normal years is somewhat corrected during heatwave years. The evaporation bias shift shows a different pattern. For HW03 and HW10, the heatwaves aggravate the overestimation of evaporation, which is coherent with the excess simulation of leaves in the model and the corresponding non-radiative cooling that they would cause. The situation in HW18b also shows

a positive shift of the same order of magnitude, but in this case it goes from a large underestimation of evaporation to a somewhat milder underestimation, which is consistent with the fact that there is less of a LAI bias. The likely explanation for this contrasting behaviour may lie in the strength of the soil-moisture/temperature coupling, which is high for HW03 and HW10 but less important for HW18b (Liu et al., 2020), and this in turn depends on differences in land cover and background climate. Croplands and grasslands dominating HW03 and HW10 deplete soil moisture more readily than forests in HW18b,

thereby triggering a more rapid release of sensible fluxes, while forests can tolerate heatwaves better thanks to deeper roots and the fact the in these northern latitudes of HW18b, the soil moisture evaporation that is lower.

The stark difference is HW18a, which one would assume would behave more like HW10 and HW03,  and that the overestimation of leaves by ERA5L leads to more simulated evaporation which in turn leads to a colder bias. Instead, the evaporation bias shifts in the other direction, going from no underestimation  to a strong underestimation, and yet a cooling LST bias





shift is also observed. This may be linked to uncertainties in the GLEAM product, which is here considered as the reference observations. GLEAM does not directly measure evaporation, but rather infers it from the data based on several modelling assumptions. Compared to flux tower estimations, GLEAM was also shown to underestimate transpiration more than ERA5-Land (Muñoz-Sabater et al., 2021). Therefore, the discrepancy in HW18a may require more investigation based on other reference sources.

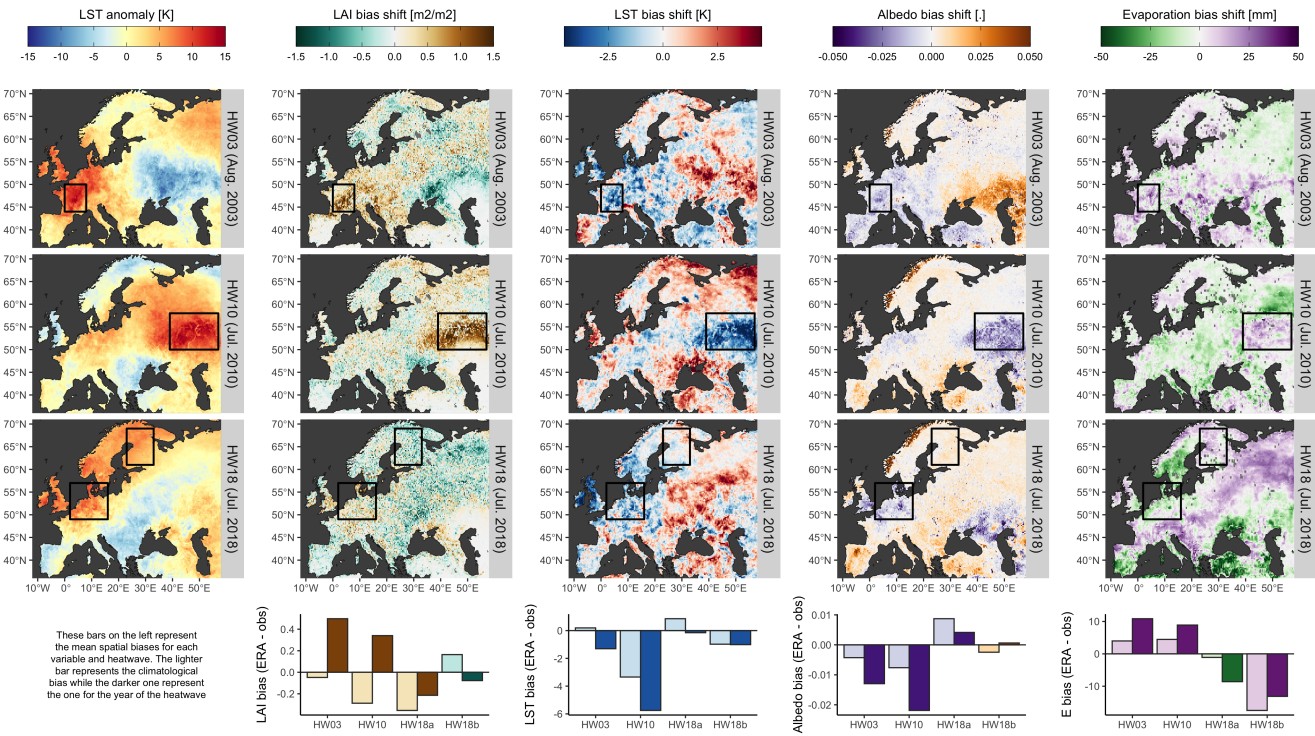

**Figure 8.** Maps of bias shifts for different variables when comparing ERA5L to what is considered here as observations. The bias shift consists of differences in the biases between the year of the heatwave and the average bias for the same period. Below the maps, we show the actual biases for each variable for the average climatological bars (light bar) and for the year of the heatwave (dark bar). The colour of the bars represents the direction of the bias shift.

**4    Discussion**

The present study proposes a novel diagnostic for land surface models centred around the key variable of LST. In the particular case of our evaluation of ERA5L, the analysis reveals the magnitude of the LST bias and its strong but heterogenous co-variation with spatio-temporal biases in LAI. It further demonstrates that these have even stronger consequences for heatwaves, when the bias in LST caused by the mis-representation of LAI is often exacerbated. The main outcome therefore is a general





warning for users of both ERA5 and ERA5-Land that the magnitude of heatwaves in particular may be underestimated in these datasets. A secondary caveat is that these datasets should not be used to assess the sensitivity of LAI to temperature, nor to other variables related to the surface energy balance, as there is a clear disconnection between both.

   LST is particularly suited to assess how models represent land-atmosphere interaction as it summarises an equilibrium point of the energy balance that can be easily observed from observations. Other variables of the energy balance, such as the latent

heat flux, cannot be captured so directly by observations, requiring instead several modelling steps along with their associated assumptions. Large discrepancies between observed and simulated LST are a strong indicator that there is a problem regarding how energy is partitioned in the model, which will further generate uncertainty in the representation of the atmosphere. In our case, there was a known suspect for the problem: the mis-representation of the inter-annual variability of LAI phenology in the ERA5L setup. However, the diagnostic we propose could easily be generalised to other state variables determining the energy

partitioning. The effect of the static representation of land cover in ERA5L could be a first example. Although, for the purposes of this study, we considered that the changes in LAI would implicitly incorporate changes in land cover, there are further layers of subtlety to be evaluated. Indeed, the distinction between different classes of low and high vegetation can have the same LAI but with different clumping patterns, resulting in different roughness lengths, which themselves could have different effects on LST.

Several discussion points can be raised with regards to extending or improving the present framework for model evaluation. A first aspect relates to the clear-sky bias in the satellite remote sensing data. Our approach to focus on the subset of days within the month that have the highest values in ERA5L should generally be robust to ensure comparability with the highest values of LST measured by the satellites, especially during the warmer season when clear skies are directly associated with higher temperatures. This max LST metric may be less effective or appropriate in wintertime, as this assumption may not always

hold: clear-sky days may sometimes be colder than overcast days. Because the measurements are done in the early afternoon when radiation load is high, it is still reasonable to believe that radiation will be the main driver determining skin temperature (rather than air temperature) in many cases, but arguably the assumption may not be as strong in winter than in summer. A possible improvement could be to work with daily values and select explicitly days in ERA5L that have clear-sky conditions. Working at a daily scale would have the added benefit of being able to isolate the effects of heatwaves more precisely than

with the monthly scale used here. Replacing MODIS LST observations with LST from geostationary satellite data, such as SEVERI on-board of MSG, could even allow pushing further by sampling from different parts of the diurnal cycle augmenting the chances of clear-sky observations. However, in all cases there is still the complication that matching clear-sky observations with the daily (or sub-daily) modelled clear-sky simulations is hampered by the model's capacity to correctly model clear-sky conditions, which could arguably be affected by the misrepresentation of LAI, introducing some kind of circularity.

Another point where the current model diagnostic could be improved relates to the associated variables used to help interpret the mechanistic interpretation of the discrepancies between LAI and LST. In the present work, we limited ourselves to albedo from MODIS and evaporation from GLEAM, and only for the heat wave analysis in the summer. First, their use as explanatory variables could be extended beyond the summer period. This was currently not done because the values provided in ERA5L only represent snow-free albedo, which do not reflect the same reality as the MODIS albedo covering all conditions. Second, the





GLEAM v3.3b used here relies on vegetation optical depth (VOD) to characterise vegetation growth, a variable that is sensitive to humidity conditions and which may thus not always be comparable with the LAI signal estimated from optical instruments. This may partially explain the inconsistencies in what is happening in HW18a, as the regions of Northern Germany and Denmark that witnessed that specific event are more humid in general than the areas in France and Russia where the other heat waves occurred. Third, other types of such diagnostic variables could be used. A prime candidate could be soil moisture

itself (SM), estimated from microwave remote sensing. In our case, we declined from using it because the corresponding C3S project had many spatial gaps (especially for year 2003) that complicated their interpretation when comparing it to the other variables.

Despite the strong discrepancies in terms of LAI and LST biases that we present in this study, it is important to point out that ERA5-Land and ERA5 remain invaluable assets for the field. They remain for many the best tools to describe many

meteorological state variables in a consistent way at hourly scale since the 1950s. We certainly continue to encourage their use. However, we should stress that the biases we expose in our results indicate that some diagnostics based on the relationships between variables in ERA5-Land and ERA5, such as assessing the sensitivity of LAI to temperature, should probably not be done as it could lead to wrong assessments.

## 5    Conclusions

The present work provides a new perspective on the importance for land surface modelling schemes to capture the dynamical nature of the interface between vegetation and atmosphere. Basically, getting the leaves right matter. Biases in LAI, which integrate this relationship between surface and atmosphere, are shown to be strongly correlated to discrepancies in the representation of surface temperature within the modelling framework behind the highly used ERA5 and ERA5-Land meteorological reanalysis datasets. The impact of not simulating dynamically the LAI cycle is more acutely demonstrated by focusing the

particular case of heat waves in Europe, where we show how their magnitude in terms of LST may be considerably underestimated. By characterising and mapping the interplay between these LAI and LST biases, our work may help users of these reanalysis datasets to anticipate where and when larger uncertainties could be expected. It should also help model developers to improve their current modelling setups by establishing a performance benchmark, and by pinpointing where and when the larger biases occur.

Overall, ECMWF analyses and reanalyses will continues to pursue the benefits of coupled data assimilation (de Rosnay et al., 2022), but the availability of stand-alone land analyses methods (Fairbairn et al., 2019) permit to examine the impact of assimilating LAI (and other land climate data records datasets) to further reduce the LST biases in future dedicated land reanalyses. Ultimately, our work does provides a strong argument to push for the assimilation of land surface variables that can be measured from satellite Earth observation, such as LAI and LST, in the weather forecasting system of ECMWF. Finally, in a

more generic conclusion reaching beyond the ECMWF system, this study could provide a blueprint for a wider benchmarking framework for land surface model evaluation that exploits the capacity of LST to integrate effects of both radiative and non-radiative processes affecting the surface energy balance.





*Code and data availability.* The code necessary to reproduce this analysis is available in a Zenodo repository: https://doi.org/10.5281/zenodo.7275088. The input data for this work is available in another dedicated Zenodo repository: https://doi.org/10.5281/zenodo.6976942

*Author contributions.* GD, MP and AC designed the study. MP gathered and preprocessed the data. GD and MP made the analyses and the figures. GD prepared the manuscript with contributions from all co-authors.

*Competing interests.* The authors declare that they have no conflict of interest.

*Acknowledgements.* Gregory Duveiller acknowledges funding by the European Research Council (ERC) Synergy Grant "Understanding and modeling the Earth System with Machine Learning (USMILE)" under the Horizon 2020 research and innovation programme (Grant 450 agreement No. 855187). The GEOV2/AVHRR LAI product was generated by CNES in the framework of the Theia land data centre, a French national inter-agency organization. The GEOV2/AVHRR algorithm was developed by CREAF and INRAE. The research leading to the current version of the product has received initial funding from various European Commission Research and Technical Development programs. The product is based on AVHRR 1km data ((c) NOAA) and is distributed by Theia.





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
