# Peer review of "Getting the leaves right matters for estimating temperature extremes"

_Geoscientific Model Development, 2022_

## Author Response (AR1)

**Response to the reviews of manuscript gmd-2022-216**

This document is a detailed point-by-point response to all referee comments where we also specify all changes in the revised manuscript. The document is structured by first indicating (1) This the comments from referees in blue, (2) author's response in black, and the (3) author's changes in manuscript highlighted in yellow.

**Responses to reviewer 1**

This paper shows that the ERA5-Land product should be used with caution and that the ERA5-Land production chain should use a more modern approach for representing vegetation or include satellite-derived LAI into their LSM. I am not sure this paper has much practical value from a modelling point of view. "Getting the leaves right matters" is perfectly right but far from being a new requirement. The added-value of this work should be better explained. Interesting recommendations are given in the Discussion section. For example, the recommendation to use hourly LST data from geostationary satellites. It is well known that solutions to integrate LAI into LSMs do exist. They are not mentioned and not used in ERA5-land. Could ERA-Land incorporate interactive LAI at some stage? Assimilation of LAI observations? Why not using another more advanced LSM forced by ERA5 atmospheric variables? The joint use of LAI and LST is interesting and offers a good benchmarking framework for assessing model performance. However, this paper may give the wrong impression that LST biases are completely explained by LAI. Other factors include the absence of representation of irrigation, snow misrepresentation, altitude solar radiation bias in mountainous areas, and slope effects in complex terrain. Overall, the paper is well written and a few changes could be sufficient to address my remarks.

We thank the reviewer for the comments and recommendations. We will try to take them fully into account in the revised manuscript. We tend to disagree to the comment that this work has little practical value from a modelling perspective. We see it as a study providing a new diagnostic of how to assess model performance (in-line with suggestions of topics from GMD). As mentioned by the reviewer, we agree that "the joint use of LAI and LST is interesting and offers a good benchmarking framework for assessing model performance." We believe the work should be judged in this light.

Following the present reviewer's suggestions, we can elaborate on why LAI is not assimilated in the ERA5 model in the revised text. One of the main strengths of the ERA5 reanalysis is its consistency and temporal depth, going back to the 1940s. If LAI would be assimilated, it would have to be done all along the archive, which is not possible because we do not have any consistent observations of LAI before the satellite era.

Finally, regarding the remark that this study may give the wrong impression that LST biases are completely explained by LAI, we do agree that this could be misinterpreted and requires further clarification.

In-line with the recommendation of the first reviewer and also what is suggested by the second reviewer, we have revised how we present the work overall. We have further added more context on why LAI is not currently assimilated in ERA5 in a dedicated paragraph in the discussion. We have clarified in the discussion that LST biases are not completely explained by LAI and mentioned the other factors brought up by the reviewer (i.e. absence of representation of irrigation, snow misrepresentation, altitude solar radiation bias in mountainous areas, and slope effects in complex terrain).

L. 40: I don't understand well the LAI definition used by the authors: "LAI is defined as half of the total green leaf area per unit horizontal ground surface area". Why half? Because only one side of the leaf is counted? For the sake of clarity, I would recommend using this more precise definition: "LAI is the one-sided green leaf area per unit horizontal ground surface area".

The reason for this more convoluted formulation of the LAI definition is that it aims at encompassing all types of leaves and not just the more typical flat leaves. As explained in Yao et al. (2019), the simpler definition employed by the reviewer is akin to that by Watson (1947): "leaf area (one side) per unit area of land". While this definition is suitable for flat leaves, it is more problematic for curly, concave or needle-shaped leaves. For non-flat leaves, such as conifer needles which are generally tubular, there is a distinction between half of total leaf area (i.e. half the surface of the tube) and the intercepted area (the cross-section of the tube). This is not a problem for flat leaves because these two concepts are the same, but the can be very different for non-flat leaves. This is why the more awkward definition tries to compensate for this by referencing everything to the total leaf area, which is proportional to the number of stomata for instance.

We have added a phrase in the text to clarify this point.

- L. 152 (GEOV2/AVHRR): The THEIA LAI data portal web page should be given. Not only the LTDR web page.

Thank you for pointing out this out. We assumed the THEIA LAI would be referred to by the citation, but indeed it seems inappropriate to provide the link to the underlying LTDR dataset more explicitly than the THEIA. We

We have now added clearly the link to were the THEIA data can be downloaded in the revised version of our manuscript.

- L. 202 (GLEAM): Which satellite date are used in this version of GLEAM? Is LST used for example?

Regarding the satellite inputs of GLEAM, the reference Martens et al. 2017 provide the list of inputs (in table 1 therein), which includes Soil Moisture, Radiation, VOD and air temperature, but it does not include LST.

We have mentioned LST is not used as an input in GLEAM.

- L. 241 ("darker than"): I am not sure this can be considered as a general rule. At wintertime, wet soils might be darker than senescent vegetation. Is this represented in the model? Adding a reference showing what is occurring in the real world would be useful.

We agree that wet soils could indeed be darker than vegetation, especially if this vegetation is senescent. Overall, we did not want to go into too much detail about this question of when and where the vegetation is darker than the background, which is why we wrote that "generally" it is darker. The comment we made refers more specifically to the fact that we see that on the one hand, in winter, the model always simulates more leaves, and on the other, we see a warm bias in the LST for Northern areas that are very likely covered with snow at some point of the winter. In such cases (i.e. too much simulated vegetation above snow), it is very likely that this would lead to a warmer surface. It is useful to note that the albedo of the model in snow-free conditions is based on a climatology of MODIS albedo observations, in-sync with the climatological LAI. It will therefore "see" if a soil is overall darker because it is generally (climatologically) wet, but it will not be sensitive to dynamic changes in soil moisture. However, the changes in albedo due to changes in snow are "prognostic", i.e. it changes along with snow cover as modelled in the re-analysis system (i.e. in-sync with dynamic weather).

We have changed the text in this part to be more explicit that we refer more specifically to the case of vegetation over snow (which will undoubtedly be darker) to avoid confusion with the case of dark moist soils. Additionally, we have added in the albedo data description section more information regarding the difference in ERA5 between the static nature of snow-free albedo and the prognostic albedo change due to snow cover albedo.

- L. 266 (Fig. 4): Figure 4 interpretation is not straightforward. This Figure does not show much more than Fig. 3 and has too many tiny sub-figures, difficult to read. The complete Figure could be moved to a Supplement and a selection of meaningful sub-figures could be left in the paper. Why not plotting the two HI and BD indices in a SM - Temperature space instead? I.e. replace Fig. 4 by the two left sub-figures of Fig. 5?

We agree that Figure 4 is a bit complex. We liked the idea of having it to illustrate the variability of situations that could be encountered, but we take well the recommendation of perhaps moving it to supplementary material. We could contemplate adding 1 or 2 more sub-figures to Figure 3 to illustrate more cases, and leave the full Fig 4 in the annex. Regarding the point of splitting Fig 5 and use half to replace Fig 4, we do not see that as necessary.

We have opted to remove Fig 4 and simply replace it by the current Fig 5.

- L. 311 (Fig. 6): Connecting maps to the colour scale is difficult. I suggest plotting only 3 colour classes: one for significant positive correlation, one for significant negative correlation (significant meaning p-value < 0.01), and white for non-significant correlation.

We have the impression that connecting the map and colour scale is not really the problem in itself, as the colourscale is clearly divergent both in colour and intensity, but rather that values based on small sample numbers might not be significant and therefore should not be shown. This was also raised by reviewer 2.

We have therefore revisited the plot as suggested by both reviewers to show only meaningful values. We have opted to mask out values based on more than 10 years and having p-values < 0.05. We kept the continuous colour scale instead of making 3 classes, as it allows for a little more information to be conveyed.

- L. 324 (Fig. 7): The top Cfb subfigure is not readable (dark green is difficult to distinguish from the dark gray color used for ocean surfaces).
Citation: https://doi.org/10.5194/gmd-2022-216-RC1

L324: We agree that the contrast is not ideal. We would want to keep the original colours of the Koppen-Geiger classification (which are widely used) and therefore we propose to adjust the background colours and thereby improve the contrast.

We have thus changed to clear background for the ocean in the revised figure.

Responses to Reviewer 2

The manuscript aims at assessing the impact of LAI errors and missing inter-annual variability in the ERA5-Land reanalysis land surface temperature (LST). This is a very relevant study for users of the reanalysis, but more generally to model development as it clearly identifies several limitations of the reanalysis LST which can be associated with errors and missing variability in LAI. The methodology is very robust, in particular (i) the focus on the hysteresis behavior between LAI and LST biases and (ii) the focus on several recent European

heatwaves. However, there are some points in the methodology and results discussion that could be clarified and are listed in the following points.

Main comment:

In this study LAI is presented as a key variable that could explain some of the errors and it is also suggested that the assimilation of LAI could reduce some of the biases. I think that this is a simplification of a very complex problem, and deserves a more detailed discussion. Land surface models are a set of parameterizations with different degrees of complexity and uncertainty. In the case of LST, it depends on the turbulent exchanges and ground heat flux. There we can identify 3 key components: (i) coupling with the underlying soil temperature; (ii) aerodynamic resistance and (iii) canopy resistance. It is expected that LAI modulates both the aerodynamic resistance (e.g. increasing drag/turbulence when LAI is higher) and canopy resistance (increasing this resistance when LAI is higher). In the case of ERA5-land only the canopy resistance is modulated by LAI linearly and normalized by the minimum canopy resistance. In other models this relation is different, and other models also account for the LAI in the aerodynamic resistance (e.g. roughness lengths in SURFEX). The message that I think it would be important to be clear is that LAI is not directly used by the model, but used as a predictor of some parameters in some parameterizations. This study makes a significant advance to our knowledge by clearly showing the relation between vegetation status (via the LAI) and the LST biases, and in my opinion, suggests the need to revise both (i) the actual LAI data used (including assimilation) and (ii) the way that the model uses this data in the different parameterizations. My suggestion to the authors, if they agree, is to have some of this discussion in the paper (discussion and conclusions).

We thank the reviewer for the insightful comments and we will strive to incorporate the suggestions as best we can in the revised manuscript. We take good note that we may have given the impression that the solution lies "simply" in assimilating LAI, while this is indeed a much more complex problem. We fully agree with the reviewer that this would deserve more details in the discussion.

As suggested, we have now added a dedicated paragraph in the discussion to emphasize how LAI is not directly used by the model, but rather used as a predictor of some parameters in some parameterizations. We also included the remark from the reviewer that this study suggests the need to revise the type of LAI used and how it can be used to improve parametrization. We believe this discussion will overall enrich the general overview of the problem by the reader.

Results & discussion clarifications

Section 2.3: Selection of 5 warmest days: The selected days from MYD11A1 and ERA5-Land are the same, or where are the 5 warmest days selected independently from each dataset ? In the discussion "A possible improvement could be to work with daily values and select explicitly days in ERA5L that have clear-sky condition" suggests that the days are not the

same. Please clarify. If the days are not the same, I see that this could be a limitation in the interpretation of some of the results, as they could be affected by the clear-sky bias.

Regarding the selection of LST values from MYD11A1 and ERA5-Land described in Section 2.3, we confirm the intuition of the Reviewer that they are not the same. We thought it was clear but realize now this may be not be as straightforward to understand. The choice of selecting the 5 warmest days is based on the assumption that there would generally be at least 5 clear-sky days in the month, and that these would likely be warmer at noon as more incoming radiation would heat up the land. We agree that this will not always be the case, and that this could, in some occasions (such as overcast winters where ERA5 might have higher surface temperatures simulated under the clouds), cause misinterpretations (which is why we wrote that phrase suggesting an improvement. The reason behind this choice is pragmatic, as it easily allows to get comparable values for most cases from both datasets in an easy way. The alternative would be to only select from ERA5 the clear sky days, but this would be more computationally complicated and would also not guarantee 100% the same days would be picked, as it implies that ERA5 currently estimates the clear-sky fraction. This is why we choose to stick to our simpler and more pragmatic approach for now, but are open for improvements in further studies.

We have clarified this point in the presentation of the data, and we reminded it both in the results (see next point) and in the discussion (where it already was mentioned).

Lines near 240: "... LAI during winter, and this corresponds to an overestimation of LST by ERA5L in the northern latitudes. This relationship between the bias in LST and the bias in LAI is consistent for such energy limited situation where biophysical effects of vegetation on climate are dominated by radiative effects. In fact, the modelling framework assumes there is an excess of leaves covering the background, the former being generally darker than the latter (especially when the background is covered with snow) resulting in more heat accumulation than what would be observed in a situation with fewer leaves."

The authors suggest that the northern hemisphere winter warm bias could be related with excessive LAI. I have some concerns with this hypothesis :

1. from a previous comment, are the 5 warmest days in each month the same in MYD11A1 and ERA5-Land ? In winter with snow conditions clear-sky conditions could be colder than cloudy-sky (stronger radiative cooling)
2. The bulk snow scheme in ERA5 and ERA5-land is known to have a warm bias due to a large thermal inertia which is partially addressed with a multi-layer snow scheme (see https://agupubs.onlinelibrary.wiley.com/doi/full/10.1029/2019MS001725)
3. The results in Figure 3b also show a rather constant LAI bias in winter with a wide range of LST biases.

Therefore I suggest the authors revise these arguments regarding the warm bias of LST in winter in the northern hemisphere, also in lines 260:265 (discussion of Figure 3b).

We take note of the remark related to the part around lines 240, as well as in lines 260-265, regarding the interpretation of the warm bias of LST in winter conditions. We confirm that the question of the 5 warmest days may indeed complicate the interpretation of winter snowy clear-sky conditions.

==We have revised the text to add caution on this interpretation and we have explicitly stated (in the results) that the pragmatic but simplified approach to map satellite LST to reanalysis can be problematic in these specific conditions.==

Figure 6 : inter-annual correlations: From my understanding the correlations were computed for each point using 16 values (years). With this sampling a correlation of 0.5 will have a p-value near 0.05. Therefore only correlations above 0.5 (simple approximation here) should be considered to avoid miss-interpretation due to the very small samples. I recommend to show this in the figure, e.g. shading non-significant correlations, and treat it in the discussion with caution.

Regarding Figure 6, we agree that given the small sample size there is ground to be more prudent and shade out those where p-values are not appropriate. This is also in-line with the remark of Reviewer 1.

==We have redone the figure masking out values that are based on 10 years or less and which have a p-value above 0.05.==

Line 307: "In other words, the years when the seasonally prescribed LAI of ERA5L is further from the reality, e.g. in years where the LAI peak is lower or shifted due to particular growing conditions of that year, the underestimation of LST can be expected to be more severe." I recommend caution on the interpretation of these results. The correlations show that in years with negative anomalies of LAI ERA5-Land has a positive LST bias. The LST warm bias can be due to the missing LAI anomaly, but such LAI anomalies are also likely reflected on other aspects of the land surface, e.g. drier soil moisture, changes in albedo, changes in surface roughness. But these results are very relevant, as they show that ERA5L LST has larger errors during extremes, which are key in our understanding of the present and future climate.

We take the Reviewer's point that some caution is warranted in this interpretation.

==We have now elaborated by stating how the anomalies in LST may be due to LAI underestimations, but also by the other effects mentioned by the reviewer (drier soil moisture, changes in albedo, changes in surface roughness).==

Results and discussion in Figure 7: By mixing all seasons I find it hard to interpret the results. Considering the case studies focus in summer, would it be easier to interpret these results if only summer months were used ? I leave this to the author's consideration.

We forgot to specify that in Figure 7 we actually only consider data from the month of August, precisely for this reason.

We have now rectified this and clearly specify in the text, the figure title and caption that the data only refer to the month of August.

375: "warning for users of both ERA5 and ERA5-Land that the magnitude of heatwaves in particular may be underestimated in these datasets" This is the case if a user uses LST to define/evaluate the heatwave event. However the most common approach in the literature is to evaluate 2-meters temperature. Please clarify here that these results only apply to Land surface temperature.

We agree.

We have now rephrased this part as follows: The main outcome of this study is therefore a general warning for users of both ERA5 and ERA5-Land about the possible shortcomings these datasets may have under heatwave conditions. Furthermore, if heatwaves were to be defined based on the skin temperature using these datasets, their magnitude would be seriously underestimated.

376: "assess the sensitivity of LAI to temperature, not to…" To avoid misunderstanding, I suggest to change to "assess the sensitivity of LAI to land surface temperature, not to"

We agree

We have implemented the change

Line:408 " This was currently not done because the values provided in ERA5L only represent snow-free albedo, which do not reflect the same reality as the MODIS albedo covering all conditions." A clean way to compute ERA5L albedo and compare with MODIS would be to compute it from the surface net solar radiation and Surface solar radiation downwards, which are available in the CDS for ERA5-Land. I don't think it would make a large difference and be necessary to be done for this study, but could be mentioned in the discussion.

We appreciate the suggestion.

We have now mentioned it in the discussion as a possible improvement